# On the Theoretical Properties of Noise Correlation in Stochastic Optimization

**Aurelien Lucchi**[*]
Department of Mathematics & Computer Science
University of Basel, Basel, Switzerland.

**Frank Proske**[*]
Department of Mathematics
University of Oslo, Oslo, Norway.

**Antonio Orvieto**
Department of Computer Science
ETH Zürich, Zürich, Switzerland.

**Francis Bach**
Inria, Ecole Normale Supérieure
PSL Research University, Paris, France.

**Hans Kersting**
Inria, Ecole Normale Supérieure
PSL Research University, Paris, France.

## Abstract

Studying the properties of stochastic noise to optimize complex non-convex functions has been an active area of research in the field of machine learning. Prior work [55, 50] has shown that the noise of stochastic gradient descent improves optimization by overcoming undesirable obstacles in the landscape. Moreover, injecting artificial Gaussian noise has become a popular idea to quickly escape saddle points. Indeed, in the absence of reliable gradient information, the noise is used to explore the landscape, but it is unclear what type of noise is optimal in terms of exploration ability. In order to narrow this gap in our knowledge, we study a general type of continuous-time non-Markovian process, based on fractional Brownian motion, that allows for the increments of the process to be correlated. This generalizes processes based on Brownian motion, such as the Ornstein-Uhlenbeck process. We demonstrate how to discretize such processes which gives rise to the new algorithm "fPGD". This method is a generalization of the known algorithms PGD and Anti-PGD [36]. We study the properties of fPGD both theoretically and empirically, demonstrating that it possesses exploration abilities that, in some cases, are favorable over PGD and Anti-PGD. These results open the field to novel ways to exploit noise for training machine learning models.

## 1 Introduction

Injecting random noise to the parameters of a function has been a commonly used technique in optimization, dating back to at least [31, 2]. More recently, [32] demonstrated the empirical benefits of injecting Gaussian noise (a.k.a. Brownian motion noise) to the gradients while training deep networks. Injecting Gaussian noise has also been a common technique used to show faster escape from saddle points [15]. Importantly, the noise of the optimizer also has an effect on the type of minima that can be reached, raising some important connections to flatness and generalization properties [24]. A distinctive property of noise is its exploration ability which is crucial when the gradient of the objective function is not reliable, either due to saddles, valleys, plateaus, or local minima. Despite a large body of work debating the distribution of the intrinsic noise of stochastic

---

[*]Shared first authorship, Correspondence: aurelien.lucchi@unibas.ch, proske@math.uio.no.

36th Conference on Neural Information Processing Systems (NeurIPS 2022).

gradient descent (SGD) [44, 54, 46, 13], there is a large gap in our knowledge regarding the role of noise to escape undesirable regions.

In order to address this question, we investigate the behavior of a general type of non-Markovian stochastic process known as fractional Brownian motion (fBM). The latter is a generalization of Brownian motion where the increments need not be independent but can instead be positively or negatively correlated. Interestingly, the processes that can be constructed from fBM recover known instances in the literature, including perturbed gradient descent (PGD) [55, 22], and anticorrelated PGD (Anti-PGD) [36, 37]. As we will see, the correlation between increments has a drastic effect of the behavior of the optimizer, and neither PGD or Anti-PGD are optimal in all cases.

**Contribution.** The first question we address is *what is the effect of the noise correlation on the overall magnitude of the process and its ability to explore the landscape?* In many applications, this is typically captured by ensuring a uniform control on the evolution of the random process. This can be achieved by bounding its expected supremum, which is connected to the concept of exit time, see Rmk. 2. To conduct such an analysis, we will consider a well-known type of continuous-time stochastic process with drift known as the fractional Ornstein-Uhlenbeck (fOU) process, which is connected to SGD (see Section 3). We will also demonstrate how to discretize such a process to obtain a practical optimization algorithm that we name fPGD (for fractional PGD). Concretely, our main result will be to provide upper and lower bounds on the expected supremum of the fOU process. These bounds will give a precise characterization of the effect of the noise correlation on the magnitude of the process. Second, we experimentally investigate the practical behavior of fBM noise injection: In Section 5.1, we demonstrate that fBM perturbations indeed help optimizers to move faster from one local minimum to another – as suggested by our theory. In Section 5.2, we demonstrate the existence of an optimization problem where non-Markovian versions of fPGD indeed outperform the known Markovian special cases (PGD and Anti-PGD) – a first proof-of-concept for the practical utility of non-Markovian noise injection.

## 2 Related work

**Non-Markovian stochastic processes.** The vast majority of the work done in stochastic optimization for machine learning focuses on Markovian processes. Yet, non-Markovian processes are known to appear naturally in many situations. They for instance describe the widths of consecutive annual rings of a tree or the values of the log return of a stock [6]. In this paper, we are especially interested in fractional Brownian Motion (fBM) as an example of a non-Markovian process. fBM is a generalization of Brownian motion where the increments need not be independent but can instead be positively or negatively correlated. The amount of correlation is captured by the Hurst parameter $H \in (0, 1)$ (a formal introduction to fBM is discussed in Section 3).

Mathematical techniques for non-Markovian processes are largely underdeveloped compared to their Markovian counterparts. There are two sets of results that are of relevance to the problem discussed in this work. The first set of results are escape times for fBM which are studied by [28, 4, 41, 45, 47], but they are restricted to simple 1-dimensional processes, and they are often applicable to specific problems such as Kramer's problem, instead of more general problems of interest. Extending these results to multiple dimensions is difficult, and we are not aware of any existing result.

The second results of relevance are bounds on the expected supremum of the stochastic process, sometimes referred to as maximal inequalities. There is a large body of work focusing on the expected supremum of "pure" fBM (i.e., without any drift term), including [34, 30, 40]. For instance, [8] establish both upper and lower bounds on the expected supremum of fBM scaling as $\mathcal{O}\left(1/\sqrt{H}\right)$. Some limited results exist with time-dependent drift terms in the 1-dimensional case, e.g. [39] gives an asymptotic result for $H \in \left(\frac{1}{2}, 1\right)$. In the multi-dimensional case, [26] derive lower and upper bound for the distribution of fBM in the multi-dimensional case, see Rmk. 4. In this work, our goal will be to provide sharp bounds for stochastic processes with more general drift terms than prior work and in the multi-dimensional case.

**Brownian motion and Ornstein-Uhlenbeck process.** A well-studied type of stochastic process is Brownian motion (BM), which is a special case of fBM (corresponding to $H = \frac{1}{2}$) where the increments are not correlated. When also considering a stochastic process with a drift, we obtain the well-known Ornstein-Uhlenbeck process, which has been the subject of a large body of work, including both results on the exit time [1, 9, 23] as well as on the expected supremum [16, 21].

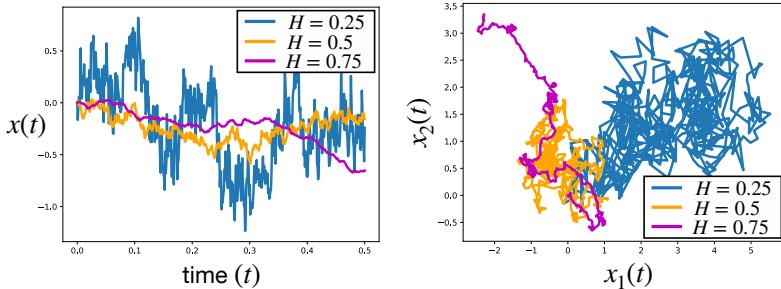

Figure 1: Discretized path of fBM in one dimension (left) and two dimensions (right) for different Hurst parameters. We observe that low values of $H$ yield stochastic processes that explore the space in a denser way that is less "biased" by the direction of the first few perturbations. This can in fact be formalized through the concept of Hausdorff dimension which is given by $d = 2 - H$ for all $H \in (0, 1)$, see Rmk. 2.

Finally, [10] also derived a maximal inequality for a multi-dimensional Ornstein-Uhlenbeck process, although their result is restricted to the radial case.

**SGD & Noise Injection.** The distribution of the noise of SGD is often a subject of debate in the literature. In some cases, the noise of SGD appears to be heavy-tailed [38, 44, 17], which yields faster exit from sharp to flat minima [33]. Noise is also often added to (stochastic) gradient descent. For instance, Perturbed Gradient Descent (PGD) is a version of GD where artificial noise is added to the parameters after every step. Multiple PGD methods have been shown to help quickly escape spurious local minima [55] and saddle points [22]. Another way to add perturbations to SGD is to add noise to the labels of the data used for training. Recent work has demonstrated that such perturbations are indeed beneficial for generalization by implicitly regularizing the loss [7, 18, 12].

## 3   Background

**Fractional Brownian Motion.** Fractional Brownian motion (fBM) is a generalization of Brownian motion where the increments of the stochastic process need not be independent. This family of Gaussian processes was developed for some hydrological modelling by [25]. [19] later studied the long term water flow characteristics of the Nile River that was affected by long periods of drought. In such a scenario, the self similarity of the process is related to a long term dependence in the water levels. Since then, fBM has also found applications in financial modeling [35]. We now give a formal definition of fBM, and discuss its main properties.

**Definition 1.** *Fractional Brownian motion (fBM) is a centered Gaussian process* $(B^H(t))_{t \in [0,T]}$ *for* $T > 0$ *with covariance function*

$$\mathbb{E}[B^H(t)B^H(s)] = \tfrac{1}{2}(|t|^{2H} + |s|^{2H} - |t - s|^{2H}),$$

*where* $H \in (0, 1)$ *is called the* ***Hurst parameter***.

Hence, fBM is a generalization of Brownian motion, which is recovered for $H = \frac{1}{2}$. If $H > \frac{1}{2}$, then the increments are positively correlated, while they are negatively correlated if $H < \frac{1}{2}$. For $H \neq \frac{1}{2}$, the increments of fBM are not independent, although they are stationary, with variance

$$\mathbb{E}(|B^H(t) - B^H(s)|^2) = |t - s|^{2H}. \tag{1}$$

A crucial property of negatively correlated increments is their ability to fill in the space, as illustrated in Fig. 1; see Rmk. 2.

**Fractional Ornstein-Uhlenbeck (OU) process.** We conduct our analysis on the fractional Ornstein-Uhlenbeck process, which can be thought of as a generalization of the well-known Ornstein-Uhlenbeck process that has already found many applications in the field of machine learning [7, 27, 48]. Formally, this process is a $d$-dimensional stochastic process $X_t^H \in \mathbb{R}^d$ defined by

$$dX_t^H = AX_t dt + dB_t^H, \quad X_0 = \xi, \tag{2}$$

where $A \in \mathbb{R}^{d \times d}$ and $B_t^H = (B_t^{H,1}, \dots B_t^{H,d})$ is a fBM with $H \in (0, 1)$. For further information regarding the fractional Ornstein-Uhlenbeck process, we refer the reader to [11, 29, 3]. We will for instance later make use of the covariance of this process that has been derived by [29, 3].

The fractional Ornstein-Uhlenbeck process is relevant for optimization purposes as it corresponds to optimizing a quadratic function of the form $f(x) = \frac{1}{2}x^\top Ax$. Indeed, the gradient of the function $\nabla f(x) = Ax$ then corresponds to the drift term in Eq. (2). We also note that when the stochastic process reaches stationarity, the stationary distribution of the iterates is known to be approximated by a quadratic function [27, 43]. This implies that such processes can be used to analyze the behavior of complex functions at stationarity, but also close to saddle points where quadratic functions often play a crucial role to analyze the escape behavior of gradient descent methods [22].

**Discretization by fractional Perturbed Gradient Descent (fPGD).** For standard Brownian motion $B_t$ (i.e. for Hurst parameter $H = 1/2$), the discretization of the continuous-time SDE $dX_t = -\nabla f(X)dt + \sigma dB_t$ of any optimization problem $\operatorname{argmin}_x f(x)$, is written

$$x_{n+1} = x_n - \eta\nabla f(x_n) + \sigma[B_{\eta(n+1)} - B_{\eta(n)}], \tag{3}$$

with step size $\eta > 0$. This method is called *perturbed gradient descent* (PGD) [22]. Here, by instead discretizing the fractional SDE $dX_t = -\nabla f(X)dt + \sigma dB_t^H$ for any $H \in (0, 1)$, we obtain a generalized version of PGD, called *fractional PGD* (fPGD):

$$x_{n+1} = x_n - \eta\nabla f(x_n) + \sigma[B_{\eta(n+1)}^H - B_{\eta(n)}^H], \tag{4}$$

which is equal to PGD if $H = 1/2$. The parameter $H$ determines the correlation of noise in fPGD: anticorrelated noise for $H \in (0, 1/2)$, positively correlated noise for $H \in (1/2, 1)$, and uncorrelated (independent) noise for $H = 1/2$. Importantly – since $B_t^H$ converges to a white-noise process (in distribution) as $H \to 0$; see Lemma 4.1 by [8] – the method Anti-PGD [36, Section 2] can be considered as the limit case of fPGD as $H \downarrow 0$. Hence, fPGD with $H \in (0, 1/2)$ interpolates between PGD ($H = 1/2$) and Anti-PGD ($H = 0$). Positive correlations ($H \in (1/2, 1)$) have not been shown to be useful yet, and indeed the below theorems and experiments will indicate that a small $H$ yields quicker escapes from difficult regions such as saddle points and local minima. While the next section will analyze the continuous-time model based on fBM, we will return to the optimizer fPGD in the experiments (Section 5) where we will illustrate how the theoretical properties relate to optimization, and why $H$ might, in some settings, be a useful tuning parameter for perturbed-gradient methods.

## 4 Analysis

### 4.1 Preliminaries

Our analysis relies on a technique known as chaining that provides a general framework that can be used to prove uniform bounds on a random process. The main tools we use are Dudley's inequality and Sudakov's inequality, which give upper and lower bounds on $\mathbb{E}\left[\sup_{t \in T} X_t\right]$ in terms of the metric entropy of $T$ denoted by $\log \mathcal{N}(T, d, \epsilon)$. Both inequalities are stated in Appendix A.1.

We will consider processes that have sub-Gaussian increments as defined below. For that, we recall the definition of the sub-Gaussian norm of a random variable $X$, denoted by $\|X\|_{\psi_2}$, as $\|X\|_{\psi_2} = \inf\{t > 0 : \mathbb{E}[\exp(X^2/t^2)] \leq 2\}$.

**Definition 2.** *A random process $(X_t)_{t \in T}$ defined on a metric space $(T, d)$ has sub-Gaussian increments if there exists $K \geq 0$ such that $\|X_t - X_s\|_{\psi_2} \leq Kd(t, s) \quad \forall t, s \in T$.*

### 4.2 One-dimensional case

In this section, we derive both upper and lower bounds on the expected supremum of the stochastic process defined in Eq. (2). Since the analysis is rather technical, we first present the main ideas in the 1-dimensional case. We then present the general result for the $d$-dimensional case.

---

**Theorem 1** (Upper bound, 1-D case)**.** *Consider the stochastic process $X_t^H$ defined by*

$$dX_t^H = \beta X_t dt + \sigma dB_t^H,$$

*with $X_0 = 0, \sigma \in \mathbb{R}$ and $|\beta| < 1$. Then, for $t \in [0, 1]$,*

$$\mathbb{E}\left[\sup_{t \in [0,1]} X_t^H\right] \leq C\left\{\sigma\sqrt{C_\beta} + \sigma\sqrt{\frac{C_\beta \pi}{H}}\right\},$$

*where $C_\beta = e^{3\beta}$ and $C$ are positive constants (independent of $H$).*

---

**Theorem 2** (Lower bound, 1-D case). *Consider the stochastic process $X_t^H$ defined by*

$$dX_t^H = -\beta X_t dt + \sigma dB_t^H,$$

*with $X_0 = 0, \sigma \in \mathbb{R}$ and $|\beta| < 1$. Then*

$$\mathbb{E}\left[\sup_{t\in[0,1]} X_t^H\right] \geq c\left\{e^{-\frac{1}{2}}\sqrt{\frac{1}{H}\log\left(\frac{\sigma C_\beta}{e^{-\frac{1}{2}}}\right)}\right\},$$

*where $C_\beta = e^{-2|\beta|}$ and $c$ are positive constants (independent of $H$).*

Note that both upper and lower bounds scale with $\mathcal{O}\left(\frac{1}{\sqrt{H}}\right)$, which implies the bound is tight and can not be improved in terms of $H$. However, the lower bound requires $\sigma > e^{2|\beta|-1/2}$ (large-noise regime) to be non-vacuous. We can also derive a lower bound for the small-noise regime as follows.

**Corollary 3** (Small noise regime). *In the case where $0 < \sigma < e^{2|\beta|-\frac{1}{2}}$, we observe that $X_t^H = \sigma Y_t^H$, where $dY_t^H = -\beta Y_t^H dt + dB_t^H, Y_0^H = 0$. Therefore*

$$\mathbb{E}\left[\sup_{t\in[0,1]} X_t^H\right] \mathbb{E}\left[\sup_{t\in[0,1]} \sigma Y_t^H\right] = \sigma\mathbb{E}\left[\sup_{t\in[0,1]} Y_t^H\right] \overset{Thm.\ 2}{\geq} \sigma c\left\{e^{-\frac{1}{2}}\sqrt{\frac{1}{H}\log\left(\frac{C_\beta}{e^{-\frac{1}{2}}}\right)}\right\}.$$

*We thus obtain a lower bound for $|\beta| \leq \frac{1}{4}$. In the general case $\frac{1}{4} < |\beta| < 1$, one gets principally a lower bound in terms of entropy numbers (Thm. 7) or by means of so called Talagrand functionals (see e.g. [45]). However, the latter quantities cannot be computed explicitly, in general. In Appendix B.1 we however provide experiments that demonstrate the $1/\sqrt{H}$ in this case, for the example of $\beta = \frac{1}{2}$.*

**Remark 1** (Extension to any finite interval). *Although the theorems are stated for the interval $T = [0,1]$, one can extend the results to an arbitrary finite interval using the self-similarity property of fBM, namely $B_{at}^H \overset{d}{=} a^H B_t^H$ for $a > 0$. This implies that $\mathbb{E}\left[\sup_{s\in[0,t]} B_s^H\right] = t^H \mathbb{E}\left[\sup_{s\in[0,1]} B_s^H\right]$. In the multi-dimensional case, the same extension works by the independence of the dimensions.*

### 4.3 Multi-dimensional case

We now generalize the bounds to the multi-dimensional case. The quantity of interest becomes the expected supremum of the norm, or more precisely $\mathbb{E}\left[\sup_{t\in[0,1]} \|X_t^H - \mathbb{E}[X_t^H]\|\right]$. Similarly to the 1-dimensional case, we apply Sudakov's inequality to the Gaussian component of the multi-dimensional fOU process $X_t^H$ [2].

**Theorem 4** (Upper bound, $d$-dimensional case). *Consider the $d$-dimensional stochastic process $X_t^H \in \mathbb{R}^d$ defined by*

$$dX_t^H = AX_t dt + dB_t^H, X_0 = \xi,$$

*where $A \in \mathbb{R}^{d\times d}$ and $B_t^H = (B_t^{H,1}, \ldots B_t^{H,d})$ is a fBM with $H \in (0,1)$. Denote by $a_{ij}(t)$ the matrix entries of $\exp(tA)A$. Then*

$$\mathbb{E}\left[\sup_{t\in[0,1]} \|X_t^H - \mathbb{E}[X_t^H]\|\right] \leq Cd^2 C_2 + \frac{Cd^2 C_2\sqrt{\pi}}{2\sqrt{H}},$$

*where $C_2 = \max_{i,j=1}^d \sup_{t\in[0,1]} |a_{ij}(t)|^2 + \max_{i,j=1}^d \sup_{t\in[0,1]} |a_{ij}(t)| + 1$ and $C > 0$ is a constant independent of $H$.*

---

[2]An alternative option, which we do not pursue in this paper, would be to generalize the result of Sudakov's inequality to the case of sub-Gaussian or sub-exponential processes. However, the proof technique presented by [49] strongly relies on Gaussian comparison inequalities and might be difficult to generalize.

**Theorem 5** (Lower bound, $d$-dimensional case). *Consider the $d$-dimensional stochastic process $X_t^H \in \mathbb{R}^d$ defined by*

$$dX_t^H = AX_t dt + dB_t^H, X_0 = \xi,$$

*where $A \in \mathbb{R}^{d \times d}$ and $B_t^H = (B_t^{H,1}, \dots B_t^{H,d})$ is a fBM with $H \in (0, \frac{1}{2})$. Denote by $a_{ij}(t)$ the matrix entries of $\exp(tA)A$ and assume that there exists $i_0 \in \{1, \dots d\}$ such that $\min_{t \in [0,1]} a_{i_0 i_0}(t) \geq -1 + C_1$, where $0 < C_1 < 1$. Then*

$$\mathbb{E}\left[ \sup_{t \in [0,1]} \|X_t^H - \mathbb{E}[X_t^H]\| \right] \geq c e^{-\frac{1}{2}} \sqrt{\frac{1}{H} \log\left(\frac{2C_1}{e^{-\frac{1}{2}}}\right)},$$

*where $c$ is a constant independent of $H$.*

We obtain an upper bound of the form $\mathcal{O}\left(\frac{1}{\sqrt{H}}\right)$ with a matching lower bound, although we do require a condition on the entries of the matrix $A$, which we discuss further in Example 1 below.

**Example 1** (Assumption lower bound). *Let $A = diag(\lambda_1, ..., \lambda_d)$. Then $e^{tA} A = diag(e^{t\lambda_1} \lambda_1, ..., e^{t\lambda_d} \lambda_d)$. If some diagonal element of $A$ is non-negative, then the assumptions of the above theorem are satisfied. Suppose that all entries $\lambda_1, ..., \lambda_d$ are negative and that $\lambda_{i_0} > -1$, for some $i_0$. Then $\min_{t \in [0,1]} a_{i_0 i_0}(t) = \min_{t \in [0,1]}(e^{t\lambda_{i_0}} \lambda_{i_0}) \geq \lambda_{i_0} > -1 + C_1$ for some $C_1 \in (0,1)$. Hence, if some diagonal element $\lambda_{i_0} > -1$ exists, then the conditions of the previous theorem are fulfilled.*

We make a few comments regarding the meaning of the bound, as well as some extensions.

First of all, the theorems discussed above bound the expected supremum of the process. We now explain how these are connected to what we call "exploration ability".

**Remark 2** (Exploration ability). *We give three different justifications for the concept of exploration. 1) Expected behavior: Based on Theorem 5, we have*

$$\mathbb{E}\left[ \sup_{t \in [0,1]} \|X_t^H\| \right] + \sup_{t \in [0,1]} \mathbb{E}\|X_t^H\| \geq \mathbb{E}\left[ \sup_{t \in [0,1]} \|X_t^H - \mathbb{E}[X_t^H]\| \right] \geq c e^{-\frac{1}{2}} \sqrt{\frac{1}{H} \log\left(\frac{2C_1}{e^{-\frac{1}{2}}}\right)},$$

*where $c, C_1$ are constant independent of $H$.*

*Since $\mathbb{E}X_t^H$ solves the ODE $\phi(t) = \xi + \int_0^t A\phi(s)ds$, the quantity $a := \sup_{t \in [0,1]} \|\mathbb{E}[X_t^H]\| < \infty$ is independent of $H$. Thus $\mathbb{E}\left[\sup_{t \in [0,1]} \|X_t^H\|\right] \geq c e^{-\frac{1}{2}} \sqrt{\frac{1}{H} \log\left(\frac{2C_1}{e^{-\frac{1}{2}}}\right)} - a$. Hence, for a given $H_1 \in \left(0, \frac{1}{2}\right)$, there is an $H < H_1$ such that $\boxed{\mathbb{E}\left[\sup_{t \in [0,1]} \|X_t^H\|\right] > \mathbb{E}\left[\sup_{t \in [0,1]} \|X_t^{H_1}\|\right]}$.*

*We conclude that the **average maximal distance of $X_t^H$ from the origin is larger than that of $X_t^{H_1}$**. This is what we mean by the process having a better exploration ability. Better exploration, in this sense, can help to escape suboptimal local minima more quickly (experiments in Section 5.1) or to explore the space around a saddle point for an exit (experiments in Section 5.2).*

*2) Exit time: First, note that $\mathbb{E}\left[\sup_{t \in [0,1]} \|X_t^H\|\right] = \int_0^\infty \mathbb{P}\left(\sup_{t \in [0,1]} \|X_t^H\| > s\right) ds$, therefore $\int_0^\infty \left\{ \mathbb{P}\left(\sup_{t \in [0,1]} \|X_t^H\| > s\right) - \mathbb{P}\left(\sup_{t \in [0,1]} \|X_t^{H_1}\| > s\right) \right\} ds > 0$.*

*Thus, we can find a $s_0 > 0$ such that $\mathbb{P}\left(\sup_{t \in [0,1]} \|X_t^H\| > s_0\right) - \mathbb{P}\left(\sup_{t \in [0,1]} \|X_t^{H_1}\| > s_0\right) > 0$ and, since probability distributions are right-continuous, there exists $\epsilon > 0$ such that for all $s \in [s_0, s_0 + \epsilon)$, $\mathbb{P}\left(\sup_{t \in [0,1]} \|X_t^H\| > s\right) > \mathbb{P}\left(\sup_{t \in [0,1]} \|X_t^{H_1}\| > s\right) > 0$. On the other hand, $\mathbb{P}\left(\sup_{t \in [0,1]} \|X_t^H\| > s\right) = \mathbb{P}(\tau_s^H < 1)$ where $\tau_s^H$ denotes the first exit time of $X^H$ from the ball $B_s(0)$ of radius $s$. We conclude that $\boxed{\mathbb{P}(\tau_s^H < 1) > \mathbb{P}(\tau_s^{H_1} < 1) \ \forall s \in [s_0, s_0 + \epsilon)}$, i.e. **the probability of an exit before time $s$ of $H$ is larger than that of $H_1$**.*

*3) Space-filling property: We note that fractional Brownian motion with low Hurst parameter $H$ fills the space more densely. This behavior comes from the fact that the graph of the fBM has a Hausdorff dimension and box-counting dimension given by $d = 2 - H$. The latter implies that $N(1/n) \approx Cn^d$ for large $n$, where $N(\varepsilon)$ denotes the number of boxes with side length $\varepsilon$, which is needed to cover the graph of the fBM. Therefore, the graph of a (1-dimensional) fBM "behaves" like a $2-$dimensional smooth manifold for small $H$. See [42]. The above claim about the Hausdorff dimension and the box-filling dimension of fBM is proved by [51]. Further related articles are [5] and [52].*

**Remark 3** (More general version). *Let us mention that the results in Theorems 4 and 5 can be also generalized to the case of linear SDE's of the form*

$$X_t = x + \int_0^t A(s)X_s ds + \int_0^t \Sigma(s)dB_s^H,$$

*where $A : [0,T] \longrightarrow \mathbb{R}^{d \times d}$ is locally bounded and measurable, $\Sigma : [0,T] \longrightarrow \mathbb{R}^{d \times d}$ is continuous differentiable and $H < \frac{1}{2}$. Here the stochastic integral with respect to $B^H$ is given by the Young integral. In fact, one can show that $\int_0^t \Sigma(s)dB_s^H = \Sigma(t)B_t^H - \int_0^t \frac{\partial}{\partial s}\Sigma(s)B_s^H ds$ (see e.g. [6]). By using the linear transformation $Y_t := X_t - \int_0^t \Sigma(s)dB_s^H$, we observe that $Y_t = x + \int_0^t (A(s)Y_s + a(s))ds$, where $a(t) := A(t) \int_0^t \Sigma(s)dB_s^H$. Denote by $\Phi(t), 0 \leq t \leq T$ the fundamental solution associated with $A(t)$, that is the unique (absolutely continuous) solution to the equation $\dot{\Phi}(t) = A(t)\Phi(t), \Phi(0) = I_{d \times d}$, the $d \times d$ identity matrix. Hence,*

$$X_t = \Phi(t)x + \Phi(t)\int_0^t \Phi^{-1}(s)A(s)\int_0^s \Sigma(u)dB_u^H ds + \int_0^t \Sigma(s)dB_s^H, 0 \leq t \leq T.$$

*Although the solution $X_t, 0 \leq t \leq T$ is in general not stationary any longer, we can use similar arguments as in the proofs of Theorems 4 and 5 and the latter representation of the process to derive a lower and upper bound for $\mathbb{E}\left[\sup_{0 \leq t \leq 1}\|X_t\|\right]$.*

**Remark 4** (Known bounds for distribution of fBM in multi-dimensional case). *In Theorems 4 and 5 we obtained a lower and upper bound for the expected supremum of the norm of a (centered) multi-dimensional fOU process (without a drift term). In comparison to the latter we also mention the following corresponding result for distributions with respect to the fBm (see Theorem 4.6 by [26]): Let $B_t^H, 0 \leq t \leq 1$ be a fractional Brownian motion in $\mathbb{R}^d$ for $H \in (0,1)$. Then the following lower and upper bound of the distribution of $\sup_{0 \leq t \leq 1}\left\|B_t^H\right\|$ ($\|\cdot\|$ maximum norm) hold: There exist constants $0 < K_1(H) \leq K_2(H) < \infty$ such that for all $0 < \varepsilon \leq 1$:*

$$e^{-dK_2(H)\varepsilon^{-\frac{1}{H}}} \leq \Pr\left(\sup_{0 \leq t \leq 1}\left\|B_t^H\right\| \leq \varepsilon\right) \leq e^{-dK_1(H)\varepsilon^{-\frac{1}{H}}}.$$

**Remark 5** (Tightness of the bound). *As in the 1-D case, the bound is right in terms of $H$, which is the most important quantity for the argument made in this paper. Regarding the dimension $d$, note that the lower bound does not depend on it (and thus it is not improvable) while the upper bound might not be tight.*

## 5 Experiments

The above analysis reveals some insights into the exploration capability of a stochastic process $X_t^H$ driven by a fractional Brownian motion $B_t^H$ with Hurst parameter $H$. We have seen that the correlation between increments directly affects the expected supremum of $X_t^H$. Concretely, lower values of the Hurst parameters are expected to yield processes with better exploration abilities, as we detailed in Remark 2. To demonstrate the relevance of our theoretical findings for optimization, we run a series of experiments with the optimizer fPGD from Eq. (4).

In Section 5.1, we show that fPGD with small $H$ better explores a non-convex loss landscapes with multiple local minima better, as predicted by our theory (Rmk. 2). In Section 5.2, we will demonstrate on an embedded saddle point that different choices of $H$ might be beneficial in different optimization phases – thus justifying the introduction of fPGD as a compromise between PGD and Anti-PGD. In the Appendix B.3, we add experiments on how fPGD behaves on bi-stable optimization landscapes, where there are two adjacent minima of different desirability (also studied by [55, 53]); the results of these experiments confirm that a small $H$ yields faster escapes from local minima.

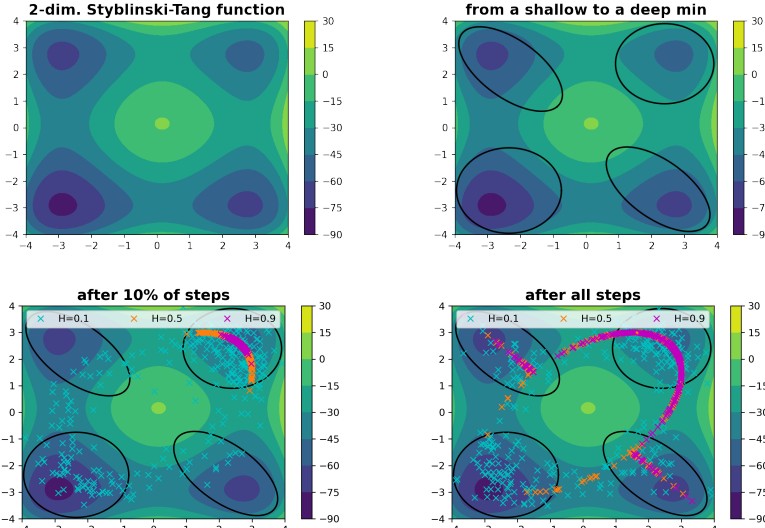

Figure 2: Exploration performance of fPGD on the Styblinski-Tang function, Eq. (5). Upper left: contour plot. Upper right: contour plot with four local minima. The local minima are roughly captured by the black ellipses. The goal is to go from the shallow (upper right) through the medium ones to the deep one (lower left). Lower right: empirical distribution of fPGD for different H after $10^4$ steps. Lower left: empirical distribution after all $10^5$ steps. (A categorization by minima is plotted in Fig. 3.) We can see that a small H deals better with this challenging escape task.

## 5.1 Demonstration of exploration ability on Styblinski–Tang function

In this section, we test the ability of fPGD to escape local minima and explore the landscape. To this end, we consider the Styblinski–Tang function in two dimensions, which is written

$$f(x, y) = \frac{1}{2} \left( x^4 - 16x^2 + 5x + y^4 - 16y^2 + 5y \right). \tag{5}$$

A contour plot of this function is depicted on the upper left of Fig. 2. This function has four minima: one shallow one (upper right), two medium ones (upper left and lower right), and one global deep minimum (lower left); see upper right subfigure of Fig. 2. Since the direct way between the shallow and deep minimum is blocked by a large hill at zero, gradient-based optimizers have to go through the medium ones, to get to the shallow one. It is therefore a suitable test problem to test the validity of the alleged exploration ability of fPGD with small $H$ (as explained in Remark 2).

We thus run experiments, where fPGD is initialized in the shallow minimum at $x_0 = (2.7, 2.7)$. After enough iterations, fPGD will eventually take the following steps: First, it will escape the shallow minimum. Second, it will reach either of the medium minima. Third, it will exit the medium minimum. Fourth, it will reach the deep minimum. (This might, of course, take a long time and the budget might not suffice.) Each of these steps can be interpreted as stopping times whose distribution we can compare for different $H$ to assess how good the exploration is: the lower each of these stopping times, the better the exploration for a given $H$.

The precise experimental setup is the following. For $H \in \{0.1, 0.3, 0.5, 0.7, 0.9\}$, we run fPGD $10^3$ times with a budget of $10^5$ steps. We chose a generic step size of $\eta = 0.01$. To zoom in on the effect, the perturbation scale $\sigma$ is set to the highest level of noise admissible without divergence on this landscape, i.e. $\sigma = 1.25$. (Note that we here again re-scale the final time to $t = 1$ so that the final amount of variance is equal for all Hurst parameters.) For each run, we determined after each step if fPGD is in the shallow, one of the medium or the deep minimum (or outside of all of them). From this information, we extracted the below summary statistics to study the exploration ability.

As a start, the lower row of Fig. 2 shows for $H \in \{0.1, 0.5, 0.9\}$, where fPGD ends up after $10^4$ and after all $10^5$ steps. It is clearly visible that the smaller the Hurst parameter, the faster fPGD moves to better local minima. (For other values of $H$ the same tendency would be visible, as the below figures will confirm.)

To give a more quantitative assessment of this effect, we provide a bar chart of the position of fPGD for all $H$ after $10^4$ and $10^5$ steps in the upper row of Fig. 3. It clearly shows that a small value of $H$ is less likely to remain in the shallow minimum and is more likely to end up in the deep minimum. This is a strong effect: after 10% of steps only $H \in \{0.1, 0.3\}$ have exited at all. Only $H = 0.1$ has a chance of reaching the deep minimum by then. While $H \in \{0.3, 0.5\}$ can eventually reach the deep minimum after an order of magnitude longer, $H \in \{0.7, 0.9\}$ cannot reach it at all in $10^5$ steps. The beneficial exploration of $H$ is confirmed by the shape of the cumulative distribution functions (cdfs) in the lower row of Fig. 3.

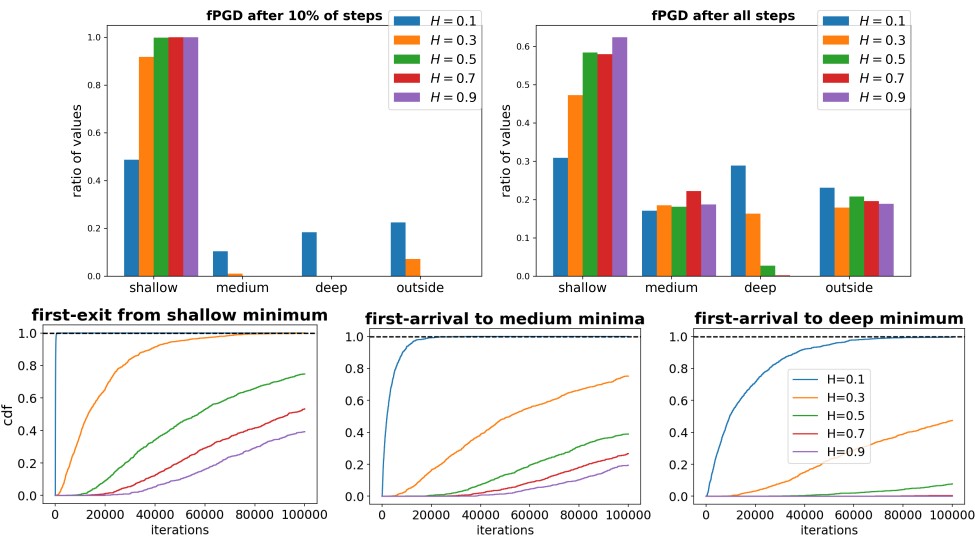

Figure 3: *Upper row:* Bar graph showing where fPGD ends up after $10^4$ (left) as well as all $10^5$ steps (right). We consider the endpoints to be in a local minimum if it ends up within the respective black ellipses (on the upper right of Fig. 2); it can end up in the shallow, one of the medium, the deep, or outside of these minima. We can see that it is much more likely for a small $H$ to reach the deep minimum. *Lower row:* Cumulative density functions (cdfs) for first exits from shallow/medium minima and first arrival to deep minimum. The upper left plot with only 10% computational budget highlights the potential for large savings in case of a small $H$. Conversely, a large $H$ is not expected to leave the shallow minimum at all (the cdf for $H = 0.9$ is smaller than 0.5 in lower left plot).

## 5.2 Non-Markovian fPGD as a trade-off between PGD and Anti-PGD

As discussed earlier, fPGD interpolates between two known Markovian methods: PGD ($H = 1/2$) and Anti-PGD ($H = 0$). Both PGD and Anti-PGD have unique advantages; see [55] or [22] for PGD and [36] for Anti-PGD. It is thus natural to ask whether there are cases where non-Markovian fPGD ($H \in (0, 1/2)$) is beneficial. While we do not have a complete answer to this question, we here present a setting that demonstrates that $H \in (0, 1/2)$ can indeed outperform both PGD ($H = 1/2$) and Anti-PGD ($H = 0$) simultaneously. We consider the following regularized quadratic: $f(x) = \frac{1}{2}x^\intercal M x + \lambda \sum_{i=1}^{d} x_i^4$, where $x_i$ is the i-th coordinate of $x \in \mathbb{R}^d$, $M \in \mathbb{R}^{d \times d}$, $d = 400$, and $\lambda = 0.0001$. If $M$ has negative eigenvalues, this landscape has a $d$-dim. saddle point at $x = 0$. If a stochastic optimizer can find the negative eigenvectors, it will reach a global minimum away from zero – while gradient descent remains stuck at zero. The problem gets increasingly hard as the number of negative eigendirections decreases [22]. Indeed, [14] showed that vanilla gradient descent takes in the worst case exponential time (in the number of dimensions) to escape such saddle points. We choose $M$ as a diagonal matrix whose diagonal entries are uniformly sampled from $[0, 1]$, and – to make it a challenging saddle – we multiply the 10 lowest of these entries by $-1$.

We test fPGD on this problem. We initialise at the saddle $x = 0$ and run fPGD with $\sigma = 0.005$ and $\eta = 0.005$ for $10^6$ iterations with different Hurst parameters. (While in general $\sigma$ and $H$ are tuning parameters that can be jointly tuned, here $\sigma = 0.005$ is a good choice for all values of $H$ which we demonstrate in Appendix B.2.) The results are presented in Fig. 4. In this problem, there is indeed a trade-off between uncorrelated ($H = 0.5$) and anticorrelated perturbations ($H < 0.5$): While a large Hurst parameter ($H = 0.5$) moves away from the saddle quickly, the non-Markovian fPGD

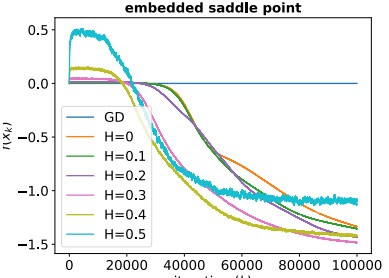
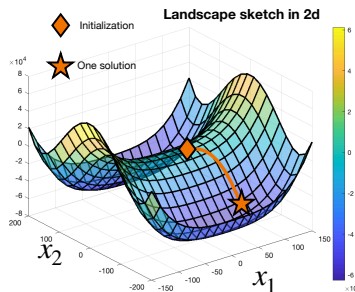

Figure 4: In the embedded saddle landscape, we see a trade-off between first exiting the saddle quickly (with a larger $H$) and then converging to the bottom of the minimum (with a smaller $H$). We plot the average over 5 runs. Non-Markovian versions of fPGD (with $H = 0.3$ or $H = 0.4$) perform better than both Markovian versions, PGD ($H = 0.5$) and Anti-PGD ($H = 0$). We use $\sigma = 0.005$ for all $H$, as justified in Appendix B.2.

($H = 0.4$) finds the direction with negative curvature most quickly (in the sense that it gets below zero). Once they have exited, all $H$ perform similarly for a while. Eventually, however, smaller Hurst parameters converge better to the bottom of the final minimum – thereby eventually catching up and surpassing the at first faster higher Hurst parameters. Which $H$ is best, will therefore depend on how many iterations the computational budget allows for. Here, the non-Markovian $H = 0.3$ and $H = 0.4$ yield the best final accuracy.

This suggests that, if all other parameters are the same, different choices of Hurst parameters are beneficial in different situations. It can thus be advantageous to tune and adapt the correlation of noise by choosing $H$. It is especially interesting that both PGD ($H = 1/2$) and Anti-PGD ($H = 0$) have shortcomings in this experiment: PGD has to increase the loss significantly in the beginning before exiting the saddle and does not converge to the bottom of the minimum in the end. Anti-PGD takes a long time to exit the saddle in the beginning. In contrast, non-Markovian fPGD (with $H = 0.3$ or $H = 0.4$) stays clear of these two undesirable extremes. We stress, however, that the above experiment is a proof of concept and only shows that there *exist* landscapes where non-Markovian fPGD performs best. On many other landscapes, PGD or Anti-PGD will still be better.

To make fPGD efficient on general landscapes, further work is needed to better understand the interplay between the scale $\sigma$ and the correlation parameter $H$, which interact according to Eq. (1).

Using this knowledge, one could design an adaptive method that jointly tunes $\sigma$ and $H$ to maximize the efficiency of perturbed-gradient methods. For the above experiments, we again emphasize that our constant choice of $\sigma$ is suitable for all $H$, as justified in Appendix B.2.

## 6   Conclusion

We analyzed the behavior of a new type of stochastic noise with correlated increments. We found that the amount of correlation, captured by the Hurst parameter $H$, has a direct effect on the overall magnitude of the process and its ability to explore. This insight opens the doors to a new area of research where the noise of stochastic processes could be manipulated to speed up the convergence of an optimization method, as well as target specific minima (e.g., depending on their flatness properties). Our experimental results validate our theoretical analysis. We showed, on some non-convex potentials, that injection of fractional noise with small $H$ speeds up the escape from a suboptimal minimum to a better one. Also, we showed that there are indeed settings where non-Markovian (somewhat anti-correlated) noise outperforms the standard Markovian (uncorrelated or perfectly anti-correlated) noise. Finally, one direction of interest would be to extend the analysis to more complex objective functions, which could potentially be achieved using the comparison theorem for SDEs [20].

**Acknowledgement**   We would like to thank Nacira Agram and Bernt Øksendal for some helpful discussions in the early stage of this project, as well as the reviewers for their feedback that greatly helped us improve this manuscript. Frank Proske acknowledges the financial support of the Center for International Cooperation in Education, project No CPEA-LT-2016/10139, Norway. Hans Kersting and Francis Bach acknowledge support from the French government under the management of the Agence Nationale de la Recherche as part of the "Investissements d'avenir" program, reference ANR-19-P3IA-0001 (PRAIRIE 3IA Institute), as well as from the European Research Council (grant SEQUOIA 724063). Aurelien Lucchi acknowledges the financial support of the Swiss National Foundation, SNF grant No 207392.

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
