# OpenReview forum: "On the Theoretical Properties of Noise Correlation in Stochastic Optimization"
_NeurIPS.cc/2022/Conference — NeurIPS 2022 Accept_

### Official Review · Reviewer_XeCx · 2022-07-11

**Rating:** 6
**Confidence:** 4
**Soundness:** 3 good
**Presentation:** 4 excellent
**Contribution:** 2 fair

**Summary:**

To discuss the effect of temporally correlated noise on exploratory capacity of stochastic optimizers, the authors have conducted a theoretical study of a newly proposed optimization algorithm/model, called 'fractional PGD' (fPGD). Instead of Gaussian perturbations, fPGD perturbs each iteration according to a fractional Brownian motion, introducing temporally correlated noise. This method is motivated as a generalization of both perturbed gradient descent (PGD), and a recently proposed optimizer called 'Anti-SGD'. To investigate the potential of the optimizer to explore, upper and lower bounds are obtained for the supremum of an Ornstein-Uhlenbeck-type fractional Brownian motion, using Gaussian process theory. The bounds demonstrate higher potential for exploration with smaller Hurst parameters. The behaviour of fPGD is demonstrated using two synthetic experiments, first by traversing a one-dimensional loss landscape with two basins, and then a 400-dimensional regularized quadratic, showing that Hurst values which interpolate between PGD and Anti-SGD appear to exhibit improved performance.

**Questions:**

As far as the linearity of the drift is concerned, would similar bounds not hold assuming the drift is bounded above and below by linear functions? I can't see anywhere in the proofs of Theorems 1 and 2 where this would be an issue, but I may be mistaken. This would significantly improve the applicability of the results.

Can the experiment in Section 5.2 be extended to a more complex optimization problem, e.g. training a neural network (ideally large, but this depends on available resources) as in the Anti-SGD paper?

Does the multi-dimensional case directly generalize the one-dimensional case? If so, why include both?

**Limitations:**

There is some discussion at the end of the experiments concerning possible future work, and a few comments on the tightness of the bounds (e.g. the multivariate bound is likely not tight in the dimension). However, there is otherwise little discussion of the limitations of the work and, in particular, its lack of generality. As per above, I believe some further discussion on this is needed.

**Strengths And Weaknesses:**

The paper is well-written, clearly presented, and well-organized. Exploration in stochastic optimization is relatively under-appreciated, and I welcome new insights which provide new  I like how Anti-PGD provides a motivating factor for the choice of fractional Brownian motion.

Unfortunately, there are a few errors in the proof which do change the results stated in the main body of the text. For example, the second line of (7) forgets the factorial terms in the expansion of $e^{\beta z}$, leading to $C_\beta = 1/(1-\beta)$, when it should be $C_\beta = e^\beta$. This is much easier to get to actually, just by noting upfront that $z \leq t - s \leq 1$ and so $e^{\beta z} \leq e^\beta$. A similar error occurs in the proof of Theorem 2.

The significance of the results, especially for the ML community, is also let down by their generality. I understand that obtaining bounds for these processes is challenging already, but Remark 3 states that a time-inhomogeneous extension is feasible; why is this not the version presented? I also suspect that even greater generality is achievable; see the Questions section.

The experiments are also promising, but I feel that they could be improved. I did not find Section 5.1 to be particularly helpful (I feel this might be improved with more and sharper basins in two dimensions). Section 5.2 displays some potential for improved empirical performance, but I am concerned that this experiment has been cherry-picked. Given the simplicity of the algorithm, I don't see why this couldn't be applied to more complex problems.

Some minor comments:
- In each of the SDEs, the $X_t$ in the drift term is missing the superscript $H$.
- For consistency, it might be good to put a superscript $H$ for $x_n$, $x_{n+1}$ (e.g. $x_n^H$) in equation (4).
- Please consider removing the italicization in the remarks, as it makes them quite challenging to read.
- line 226: 'the bound is right' -> 'the bound is tight' ?

---

> ### Author Response · Authors · 2022-08-01
> **Answers**
>
> Dear reviewer, Thank you for your valuable feedback. We appreciate your time reviewing this paper. For your convenience, we uploaded an updated PDF where the changed parts are highlighted in _blue_. The new parts include a new Corollary, two new Appendices with additional experiments and multiple clarifications in the text.
>
> We are glad you recognize the topic addressed in the paper is under-explored and worth further investigation. Please note that there is no mistake in our proof. We answer your questions below and are happy to provide further feedback if needed.
>
>
> # Weaknesses
>
> 1) "Errors in the proof":
> Please note that there is no mistake in the proofs. We did not forget the factorial terms, we simply bounded them from above by a larger term. However, you are right that your alternative is simpler and leads to a tighter bound since $e^\beta \leq 1/(1-\beta)$ for $\beta \in [0,1]$. Thank you for pointing this out!
>
> ---
>
> 2) More general version:
> We did not present a more general version as it would make the notation much heavier and the estimates difficult to parse. Please note that our results already present a significant level of novelty compared to prior work. We believe that the current version strikes a good balance between readability and generality.
>
> ---
>
> 3) Experiments are promising but...
> Thanks for these great suggestions! Regarding Section 5.1., we agree that better requirements on two-dimensional landscapes (with more local minima) would be an improvement. We added them in Appendix B.3, on the Styblinski-Tang function; the code is provided in the updated Supplementary Material.
> Regarding Section 5.2., you are right that we chose the landscape consciously and that fPGD (in its current form) will not be the best method on many landscapes. The goal of this experiment is simply to demonstrate that there _exist_ landscapes where _slightly_ anti-correlated noise outperforms uncorrelated or fully anticorrelated noise. It is just a proof of concept, not a claim about general performance. We explicitly stress this in the new version (see blue part of Section 5.2).
> To make fPGD a practical method on general landscapes, future work will be necessary. The main challenge will probably be the tuning of the Hurst parameter H (or perhaps even the joint tuning of H and sigma). Different local environments will require different parameter settings (as is already clear from the different benefits of PGD and Anti-PGD).
> But this will be a significant research effort. Hence, in our opinion, our proof of concept suffices for now, to demonstrate the potential of fPGD.
>
> # Questions
>
> 1) Linearity of the drift:
> Yes, that’s indeed a valid suggestion. We focused on the (fractional) Ornstein-Uhlenbeck process due to its prevalence in prior work, including many works in the field of machine learning (e.g. Mandt et al.). The question would be whether drift terms bounded above and below by linear functions are relevant in practice. Can you please comment on whether that is the case?
>
> ---
>
> 2) Extending experiments in section 5.2:
> In principle yes! We haven’t done so yet because our focus so far was on developing a theoretical framework to demonstrate the benefit of fBM. There is a lot more that can be developed as future work. To make the use of fBM practical in complex landscapes such as the ones of deep neural networks, one might first need to develop a procedure to automatically choose the Hurst parameter H, therefore requiring more engineering effort. This is certainly a direction we are planning to investigate. Yet, we think our submission is worth publishing as is, since it already presents a significant degree of novelty, with both theory and practical experiments (albeit not complex neural networks).
> Also note that we added additional experiments on the Styblinski-Tang function in Appendix B.3 (as we pointed out above).
>
> ---
>
> 3) "Does the multi-dimensional case directly generalize the one-dimensional case?"
> We could indeed have stated the multi-dimensional case first and then derive a result for the one-dimensional case. We chose to do the reverse for two reasons:
> a) We would like to emphasize that the bounds in the one-dimensional case is novel so we thought it would be worth presenting it separately.
> b) The proof of the one-dimensional case is simpler, which we believe increases the didactic value of the paper. As stated in the paper, there has been a lot of prior work studying fBM in the 1-dimensional case, so it also makes it easier to compare to them.

---

> > ### Comment · Reviewer_XeCx · 2022-08-08
> > **response**
> >
> > Thank you to the authors for responding to my concerns, and my apologies for taking time before responding. Thank you for the clarification regarding the proof; indeed, I was incorrect to suggest the results were erroneous, but are likely much sharper now with $e^\beta$ instead of $1/(1-\beta)$. The additional experiments are excellent and demonstrate the exploratory behavior much better.
> >
> > I appreciate that the results in the paper are novel. My primary concern is that there is nothing in the document that can plausibly tie into more complex models, so discussing this work in the context of machine learning comes across as a bit disingenuous. I would be happy to recommend the paper if there was something either theoretical or empirical that suggests these ideas extend to more general settings.
> >
> > If it is difficult to show that fPGD is competitive for more complex problems, what about replicating the noise injection experiment in the Anti-PGD paper? Alternatively, on the theory side, considering drift terms bounded in magnitude by a linear function allows for arbitrary objectives with Lipschitz gradients. Even more specifically, one could consider the case where the drift is bounded above and below by linear functions as I suggested, as this allows for training models with bounded derivatives under $L^2$ regularization (optimizing $f(x) + \lambda ||x||^2$). This incorporates a very general class of models, and so these assumptions are often considered in the ML optimization literature. This would be more than enough to demonstrate the results apply more generally.
> >
> > Since there is little time before the end of the response period, I would be happy with an acknowledgement whether such a thing is possible or not, or if you have any other ideas. I think any demonstration or discussion of how these concepts might extend more generally would be valuable for this work to be more widely appreciated in the community.

---

> > > ### Author Response · Authors · 2022-08-08
> > > **Response - part 1**
> > >
> > > Dear reviewer,
> > >
> > > Thank you for your reply, this is sincerely greatly appreciated. You are right that the nature of our work is more theoretical and that it therefore has some practical limitations. We will update the manuscript to add a discussion about this aspect. Importantly, we also would like to point out the following aspects that we believe make our work of interest for the field of machine learning.
> > >
> > > ## Part 1: Experimentally
> > >
> > > First, we showcased the behavior of fPGD on a landscape with saddles. It is well-known that saddles do appear in certain conditions in machine learning. This includes for instance:
> > > * Autoencoders at initialization [1]
> > > * Certain types of neural network architectures [2]
> > > * Various reinforcement learning policy optimization problems [3]
> > > Note that all papers have been published in machine learning venues. In addition, the study of saddle points has been the subject of a large body of work in machine learning, see for instance [4],[5] and [6] among many others.
> > >
> > > [1] Kunin, Daniel, et al. "Loss landscapes of regularized linear autoencoders." International Conference on Machine Learning. PMLR, 2019.
> > > [2] Dauphin, Yann N., et al. "Identifying and attacking the saddle point problem in high-dimensional non-convex optimization." Advances in neural information processing systems 27 (2014).
> > > [3] Ahmed, Zafarali, et al. "Understanding the impact of entropy on policy optimization." International conference on machine learning. PMLR, 2019.
> > >
> > > [4] Ge, Rong, et al. "Escaping from saddle points—online stochastic gradient for tensor decomposition." Conference on learning theory. PMLR, 2015.
> > > [5] Jin, Chi, et al. "How to escape saddle points efficiently." International Conference on Machine Learning. PMLR, 2017.
> > > [6] Daneshmand, Hadi, et al. "Escaping saddles with stochastic gradients." International Conference on Machine Learning. PMLR, 2018.
> > >
> > > Second, we also showcased the experimental behavior on the Kramer's escape problem. Although this landscape does appear more artificial, it is also an important model to study in the machine learning community, and we can once again cite a long list of papers accepted in machine learning venues such as
> > > [7] Nguyen, Thanh, Umut Simsekli, Mert Gürbüzbalaban, and Gael Richard. “First Exit Time Analysis of Stochastic Gradient Descent Under Heavy-Tailed Gradient Noise”, NeurIPS 2019.
> > > [8] Xie, Zeke, Issei Sato, and Masashi Sugiyama. "A diffusion theory for deep learning dynamics: Stochastic gradient descent exponentially favors flat minima." arXiv preprint arXiv:2002.03495 (2020).
> > > [9] Liu, Kangqiao, Liu Ziyin, and Masahito Ueda. "Noise and fluctuation of finite learning rate stochastic gradient descent." International Conference on Machine Learning. PMLR, 2021.

---

> > > > ### Author Response · Authors · 2022-08-08
> > > > **Response - part 2**
> > > >
> > > > ## Part 2: Theoretically
> > > >
> > > > We study a simpler model known as the Ornstein–Uhlenbeck process, which corresponds to optimizing a quadratic function with stochastic gradient descent (gradient descent with Gaussian noise to be more exact). This type of model has once again been used in the machine learning literature, see e.g.
> > > > [10] Mandt, Stephan, Matthew Hoffman, and David Blei. "A variational analysis of stochastic gradient algorithms." International conference on machine learning. PMLR, 2016.
> > > > [11] Li, Qianxiao, Cheng Tai, and Weinan E. “Stochastic modified equations and adaptive stochastic gradient algorithms”. ICML 2017
> > > > [12] Blanc, Guy, Neha Gupta, Gregory Valiant, Paul Valiant “Implicit regularization for deep neural networks driven by an Ornstein-Uhlenbeck like process”. Conference on learning theory (COLT), 2020
> > > >
> > > > Finally, your suggestion to extend the theory is interesting and we do think some results can be produced in this direction. Given the rather short timeline, we are probably not able to write this formally but here is a somewhat informal explanation.
> > > >
> > > > First of all, the result you are asking for can be proven in the 1-dimensional case using the comparison theorem for SDEs, see the classical reference [13]:
> > > > [13] Ikeda, Nobuyuki, and Shinzo Watanabe. "A comparison theorem for solutions of stochastic differential equations and its applications." Osaka Journal of Mathematics 14.3 (1977): 619-633.
> > > > We are not aware of a multi-dimensional version of this theorem, but this is definitely something of interest and we will consider it.
> > > >
> > > > For the general multi-dimensional case, there is a simple way to extend our theoretical results to more complex models (i.e. with a non-quadratic drift term) under the following assumption: If one assumes that the sup-norm of the solution of an arbitrary (complex) process can be lower and upper bounded by the sup-norm of the process analyzed in the paper, our results will provide an upper and lower bound on the sup-norm of the complex process. Here, we do have to acknowledge that we do not know the extent to which this could be of practical relevance. If you are interested in any arbitrary complex drift term, we can use the Girsanov theorem. Basically the Girsanov theorem says that if we change the drift coefficient of a given Ito process (with a nondegenerate diffusion coefficient), then the law of the process will not change dramatically. In fact, the law of the new process will be absolutely continuous w.r.t. the law of the original process and we can compute explicitly the Radon-Nikodym derivative.
> > > >
> > > > So in summary, one can already extend the lower and upper bounds to more complex drift terms in the 1-dimensional case and the upper bound in the multi-dimensional case. For the lower bound, one needs some further assumptions, but we do think an extension to the general case might be feasible. Thank you for pointing this out. We hope that our answer addresses your comments and we are happy to discuss more if you have further questions.

---

> > > > > ### Comment · Reviewer_XeCx · 2022-08-09
> > > > > **Thank you**
> > > > >
> > > > > Thank you for the quick and detailed response. Experimentally, I recognize that investigating the behavior of certain properties exhibited by neural network landscapes in isolation is useful. However, these are typically conducted in addition to a larger experiment that considers a more complex problem. The experiments in the paper are useful; it is only a demonstration of the behavior (positive or negative) on a high-dimensional problem that would really fill in the final gaps here.
> > > > >
> > > > > If such an experiment is difficult, then the theoretical perspective is probably more worthwhile. The application of the comparison theorem in the one-dimensional case is excellent: I think this is worth mentioning, even if it is a single sentence before the discussion of the main 1D theorem. I agree that Girsanov's theorem would probably be sufficient to cover the general case. I'm guessing the argument would go something like the following:
> > > > >
> > > > > Assume that $f$ is a function with bounded gradient $\sup_x ||\nabla f(x)||_2 \leq M$. Let $X_t$ and $\tilde{X}_t$ be defined by
> > > > >
> > > > > - $dX_{t}=AX_{t}dt+dW_{t}^{H}$
> > > > > - $d\tilde{X}_{t}=-\nabla f(\tilde{X}_t) dt +A \tilde{X}_t dt+dW_t^H$.
> > > > >
> > > > > Then from Girsanov's theorem, (letting the sup of $t \leq T$ operator be denoted by $\mathcal{S}$, since this seems to break the Markdown interpreter),
> > > > >
> > > > > $\mathbb{E} \mathcal{S} || \tilde{X}_t || = \mathbb{E} [\exp(-\int_0^T \frac{1}{2} ||\nabla f(X_s)||^2 ds -\int_0^T \nabla f(X_t) . dW_t^H) \mathcal{S} || X_t ||]$
> > > > >
> > > > > and so for $I_T = \int_0^T \nabla f(X_t) \cdot dW_t^H$,
> > > > >
> > > > > $\mathbb{E}[\exp(-I_T-\frac{1}{2} M^2 T) \mathcal{S} ||X_t|| ] \leq \mathbb{E}\mathcal{S}|| \tilde{X}_t || \leq \mathbb{E}[\exp(-I_T)\mathcal{S} ||X_t||] $
> > > > >
> > > > > I'm not quite sure how the $I_T$ term can be dealt with though --- is this what you had in mind?
> > > > >
> > > > > If this sort of argument can extend the current results to allow for optimizing functions of the form $f(x) + \beta ||x||^2$ where $f$ has bounded gradient (subject to appropriately large constants), this would be a huge improvement, and would highlight the value of the results for those who are less theoretically inclined. For example, $f(x) + \beta ||x||^2$ could allow for arbitrary "funnel-shaped" objectives, or simply classes of models with $L^2$ regularization. I would imagine such an argument would belong in the Supplementary Material, but a mention of this in the main body above the results would go a long way.

---

> > > > > > ### Author Response · Authors · 2022-08-09
> > > > > > **Answer**
> > > > > >
> > > > > > Dear reviewer,
> > > > > >
> > > > > > Thank you for your feedback. We will update the manuscript to explain how to extend our results to general objective functions in the 1-dimensional case using the comparison theorem. We will also add a discussion about various ways to extend the resul in the n-dimensional case (with a discussion of the limitations we mentioned above).
> > > > > >
> > > > > > Regarding your question about Girsanov's theorem, it seems that you are not using the correct version. One in fact needs to use the existing version for fractional Brownian motion. See for instance the seminal reference by Decreusefond, Laurent. "Stochastic analysis of the fractional Brownian motion." Potential analysis 10.2 (1999): 177-214.
> > > > > >
> > > > > > Alternatively, there is a more recent version of this Theorem in https://arxiv.org/pdf/2104.14971.pdf (see Theorem 2.1). We will not obtain an expression as nice as in the quadratic case but this will give a valid upper bound on the supremum. Does this answer your question?
> > > > > >
> > > > > > Best.
> > > > > > The authors

---

> > > > > > > ### Comment · Reviewer_XeCx · 2022-08-09
> > > > > > > **Thank you for the correction**
> > > > > > >
> > > > > > > Thank you for your quick response; apologies for my error with the application of Girsanov's theorem, I had assumed it followed similarly to the Brownian motion case but forgot the covariance kernel! Thank you for linking to the correct analogue. I can see how this argument would proceed now, and while the bounds would be unpleasant by comparison, I believe this is a significant addition for extending the theory to more complex models. In addition to including this argument explicitly in the Appendix, I would recommend a few comments in the main body of the paper above the multidimensional results explaining that bounds for a much wider variety of models reduce to the linear case by using Girsanov's theorem.  I am convinced this should be enough to cover sufficiently general use cases, and have consequently updated my score.

---

> > > > > > > > ### Author Response · Authors · 2022-08-09
> > > > > > > > **Thank you**
> > > > > > > >
> > > > > > > > Dear reviewer,
> > > > > > > >
> > > > > > > > Once again, thank you for the feedback. We are glad we had what we consider to be a very productive exchange that will help us to improve our paper. We have already updated the draft and will make more updates as we promissed.
> > > > > > > >
> > > > > > > > Best,
> > > > > > > > The authors

---

> ### Author Response · Authors · 2022-08-08
> **Gentle reminder for response**
>
> Dear Reviewer,
>
> We would like to sincerely thank you for your time and efforts in reviewing our manuscript. We have provided answers to your comments and questions. As the author-reviewer discussion period is ending soon, we would like to kindly request to please let us know if you have any further concerns.
>
> Best regards,
> The authors

---

### Official Review · Reviewer_9UJ2 · 2022-07-13

**Rating:** 7
**Confidence:** 2
**Soundness:** 2 fair
**Presentation:** 3 good
**Contribution:** 3 good

**Summary:**

Generalization of gradient based algorithms for highly non-convex loss landscapes is poorly understood and there are many hypotheses explaining the empirical observations. One of them is that flat minima generalize better than sharp minima. The stochasticity of gradient based plays an important role in escaping sharp minima and settling on flat minima, and this paper studies the basic properties of fractional Ornstein-Uhlenbeck process and its dynamics on toy examples to better understand the importance of correlation in noise.

**Questions:**

1. In Remark 5, the paper says that the lower bound does not depend on $d$. I am not sure I understand since the lower bound clearly depends on $d$. Could you please explain what is meant here?
2. In equation (5), should the characteristic function in the middle term in RHS be $\chi_{a \le x \le c}$?

**Limitations:**

Not relevant

**Strengths And Weaknesses:**

The paper is clear and easy to understand. It makes nice theoretical contributions in Section-4 on giving bounds for the expected suprema, which justifies the intuition that smaller Hurst parameter corresponds to a higher exploration ability of the stochastic process. It might be useful to mention that a fractional Ornstein-Uhlenbeck process is a Gaussian process because that fact is used in the proofs of the theorems implicitly.

While the contribution of generalizing PGD and Anti-PGD is useful, I was expecting more experiments and more realistic examples to better understand the effect of $H$ strictly between $0$ and $1/2$. I was also expecting more analysis for the case $H > 1/2$, and when that could be useful if at all.

---

> ### Author Response · Authors · 2022-08-01
> **Answers**
>
> Dear reviewer, Thank you for your valuable feedback. We appreciate your time reviewing this paper. For your convenience, we uploaded an updated PDF where the changed parts are highlighted in _blue_. The new parts include a new Corollary, two new Appendices with additional experiments and multiple clarifications in the text.
>
> We are glad you recognize the novelty of the paper and we thank you for your comments. We answer your questions below.
>
> 1) The lower bound appearing in Theorem 4 does not have any direct dependency to the dimension d. This is simply due to the fact that we only require a rather mild assumption on one diagonal entry of the matrix $A$ (more precisely $\exp(At)A)$. If one wants to have a dependency on d appearing in the bound, this would logically require adding a condition on multiple entries (we for instance need to avoid the fact that one can embed a 1-dimensional process in a d-dimensional space). We will add a comment about this.
>
> 2) Yes, thank you for spotting this typo which we fixed in the new version!

---

### Official Review · Reviewer_zaJL · 2022-07-13

**Rating:** 5
**Confidence:** 4
**Soundness:** 2 fair
**Presentation:** 3 good
**Contribution:** 2 fair

**Summary:**

This paper proposes fractional perturbed gradient descent (fPGD), which corresponds to the discretization of a stochastic differential equation driven by fractional Brownian motion.
fPGD (with a parameter $H \in (0,1)$) has a nice property of interpolating two existing algorithms, i.e., PGD (with $H=0.5$) and anti-PGD (with $H = 0$).

In terms of theory, the paper provided upper and lower bounds for the expected supermum of fractional SDEs, in both one-dimension cases and multi-dim cases. Note that the expected supermum captures the escaping/exploration ability of the algorithm.

In terms of experiments, the paper shows that, at least in simulated dataset, fPGD achieves a better tradeoff between the extremes PGD and anti-PGD.


**Questions:**

1. Time and space complexity for generating the noise required by eq(4). Note that in eq(4), all noise is correlated. Suppose we need to run the algorithm for $N$ steps, i.e., we need to generate $N$ correlated noise, can you explain the time and space complexity for generating that noise? If the time/space complexity is overly large in terms of $N$, the fPGD algorithm might not be of practical interests.

2. Thm1. In fact the integral can be solved in an analytic form --- please simply the expression in this regard.

3. Thm2. For the lower bound to be non-negative, one needs to have that $\sigma > \exp(2 |\beta| - 0.5)$. This seems to limit the applicability of the theorem: the bound is trivial when noise is small. Please clarify this during the rebuttal. Can we obtain some lower bounds for the small noise regime?

4. Line 160 seems to be overclaiming: the bound might not be tight in terms of $H$, if the noise is small.

5. Remark 1. Can you clarify whether or not remark 1 applies to multi-dim cases?

6. Thms 3 and 4 assume that $X_0 = \xi$. Why not assume that $X_0 = 0$ to be consistent with Thms 1 and 2? I believe this can be assumed without loss of generality?

7. Thms 3 and 4 assume $\sigma = 1$ or $\Sigma = I$. It is not clear how a scaling/rotation on the fractional Brownian motion affects the final bounds. At least the effect of scaling (i.e., $\Sigma = \sigma I$ for general $\sigma$) should be discussed, as Thms 1 and 2.

8. Thms 3 and 4. The condition of $C_2$ or $C_1$ is hard to interpret. Could you please explain the intuition behind them?

9. Thm 3 might be too loose in terms of $d$. In fact, appendix line 606 suggests that the authors control an $\ell_2$ norm by a sum of $d$ $\ell_\infty$ norm. This inequality is too crude and loses the geometry of $\ell_2$-norm.  I am not sure how useful the upper bound is.

10. Similarly, for Thm 4 to be nontrivial, one needs to have $C_1 > 0.5 e^{-0.5}$. The comments on Example 1 need revision accordingly.

11. Line 189. I feel this sentence overclaims the benefits of "exploration". I agree that more exploration helps escape from suboptimals, but on the other hand, the whole algorithm becomes more unstable. Perhaps the authors should comment here that there is a trade-off between exploration vs. convergence (in the sense that the algorithm stays in a basin of a good minima with good probability). Similarly the authors should revise the comments on the "space-filling property".

12. Remark 3, could you comment on how the bounds are affected by considering a time-dependent $A$ and $\Sigma$?

13. Remark 4. Could you explain how this result compares to your results?


**Limitations:**

See above.


===Post-Rebuttal

Some of my concerns are addressed. The major issue is in the high-dim analysis, which does not make use of the right geometry and seems to be artificial in many aspects. I will raise my score from 4 to 5 but cannot give a higher score due to the mentioned limitations of this work.

**Strengths And Weaknesses:**

# Strengths

+ The fPGD algorithm is quite novel (and very interesting to me!), at least in the machine learning field, to my knowledge.
+ The fPGD algorithm has a nice theory background of being an discretization of a SDE driven by fractional Brownian motion.
+ FPGD interpolates two existing algorithms, and is verified in simulations that can achieve a better tradeoff than both of them.

# Weakness
- The time and space complexity of generating noise for fPGD is not discussed.
- The upper and lower bounds have multiple limitations. Please see more discussion below.
- No real machine learning experiments.

---

> ### Author Response · Authors · 2022-08-01
> **Answers: part 1**
>
> Dear reviewer,
>
> Thank you for your valuable feedback. We appreciate your time reviewing this paper. For your convenience, we uploaded an updated PDF where the changed parts are highlighted in _blue_. The new parts include a new Corollary, two new Appendices with additional experiments and multiple clarifications in the text.
>
> We are glad you recognize the novelty of the paper and we thank you for your comments. Since many of the comments are technical, we would like to emphasize that we purposefully decided to present a version of our result on a less general version in order to make the paper easier to understand for a general audience. We provide detailed answers to your comments below and we would be glad to provide further feedback if necessary.
>
> 1) Very good question. This is in fact a well-researched question where approximate methods have been proposed in the literature, including for instance the random midpoint displacement (RMD) method (by Lau et al, see reference below). RMD works by subdividing the interval recursively and constructing the values of the process at the midpoints from the values at the endpoints. The complexity depends on the number of midpoints. The original paper reports that generating  260,000 observations only takes about a couple of minutes on a SUN SPARCstation 20, which is of course a very limited machine compared to current hardware.
> Lau, Wing-Cheong, et al. "Self-similar traffic generation: The random midpoint displacement algorithm and its properties." Proceedings IEEE International Conference on Communications ICC'95. Vol. 1. IEEE, 1995.
>
> 2) We did not manage to get a closed-form for this integral by direct calculations. We are of course happy to simplify this expression if you have a suggestion!
>
> 3) In the case of $0<\sigma <e^{2\left\vert \beta \right\vert -
> \frac{1}{2}}$ we observe that $X_{t}^{H}=\sigma Y_{t}^{H}\text{,}$
> where $dY_{t}^{H}=-\beta Y_{t}^{H}dt+dB_{t}^{H},Y_{0}^{H}=0\text{.}$
> Therefore
> $E \left[ \sup_{t\in \left[ 0,1\right] }X_{t}^{H} \right] = E \left[ \sup_{t\in \left[ 0,1\right] }\sigma Y_{t}^{H}\right] = \sigma E \left[ \sup_{t\in \left[0,1\right] }Y_{t}^{H}\right]$,
> and by Theorem 2, we conclude that
> $E \left[ \sup_{t\in \left[ 0,1\right] }X_{t}^{H} \right] \geq \sigma c \left [ e^{-\frac{1}{2}}\sqrt{\frac{1}{H}\log (\frac{C_{\beta }}{e^{-\frac{1}{2}}})} \right] .$
>
> Therefore we obtain a lower bound in this case, when $\left\vert \beta \right\vert
> \leq \frac{1}{4}$. In the general case $\frac{1}{4}<\left\vert \beta
> \right\vert <1$, one gets principally a lower bound in terms of entropy
> numbers (Thm6) or by means of so called Talagrand functionals (see e.g. $\left[ \text{45}\right] $). However, the latter quantities cannot be
> computed explicitly, in general.
>
> We added this lower bound to the new version and we also added some new experimental results in Appendix B.1 that back up the bound (including for choices of $\beta$ and $\sigma$ that are not covered by our theory); the code is provided in the updated Supplementary Material.
>
> 4) Please see our answer #3 above about how to extend the range of the lower bound, as well as the new experimental results (appendix B.1) that support the bounds we derived. We are of course happy to adjust the claim but we think there is enough evidence for it, please let us know if you agree/disagree.
>
> 5) Yes, Remark 1 also applies to the multi-dimensional theorems. This is a direct consequence of the independence of the fBMs in each dimension. We updated the remark accordingly.
>
> 6) Yes, we simply wanted to have a more general version in Theorems 3 and 4. We can very easily adjust this if necessary, there is absolutely no difficulty to do so.
>
> 7) One can also derive a similar result to Thm. 3 in the case of a
> non-singular diffusion matrix $\sigma$ by using the representation (see answer #2)
> $X_{t}^{H}=\sigma Y_{t}^{H}\text{,}$ where $dY_{t}^{H}=\sigma ^{-1}A\sigma Y_{t}^{H}dt+dB_{t}^{H},Y_{0}^{H}=\sigma^{-1}\xi \text{.}$
>
> In fact, in this case we see that
> $$
> E\left[ \sup_{t\in \left[ 0,1\right] }\left\Vert X_{t}^{H}-E\left[ X_{t}^{H}
> \right] \right\Vert \right]  = E\left[ \sup_{t\in \left[ 0,1\right]
> }\left\Vert \sigma (Y_{t}^{H}-E\left[ Y_{t}^{H}\right] )\right\Vert \right]
> \leq \left\vert \sigma \right\vert E\left[ \sup_{t\in \left[ 0,1\right]
> }\left\Vert Y_{t}^{H}-E\left[ Y_{t}^{H}\right] \right\Vert \right] \text{,}
> $$
> where $\left\vert \sigma \right\vert $ is a matrix norm of $\sigma $ and
> where $Y_{t}^{H}$ is a process which Thm3 can be applied to (for $A$
> replaced by $\sigma ^{-1}A\sigma $).
>
> As for a similar lower bound to Thm4 in the case of a non-singular $\sigma $, one may follow the proof of Thm4, since the process still is stationary.
> However, the proof requires more care than in the case of $\sigma =Id$, since one has to take into account the covariance structure of the process $\sigma B_{t}^{H}$, which leads to more complicated expressions in the lower
> bound.

---

> > ### Author Response · Authors · 2022-08-01
> > **Answers: part 2**
> >
> > 8) Regarding C_2: Please note that there is no condition on C_2, it is defined in the statement of Theorem 3 (upper bound) and it is by definition positive. It enters the bound in the log term but unlike the lower bound, this is just whatever constant value it takes given the drift matrix A. We hope this clarifies this point.
> > Regarding C_1: you are right, there is a condition stated in Theorem 4 (lower bound).
> > We agree this condition is not intuitive at first glance. We gave some intuition in Example 1 (following the theorem statement). It states that the condition is for instance satisfied if at least one diagonal entry is non-negative. This is a rather mild condition. One problematic case would be when all eigenvalues are negative but this corresponds to a very degenerate case. So in brief, the condition on C_1 does not appear restrictive for machine learning applications. If Example 1 is not clear enough, please let us know what part needs to be changed and we will adjust the claim accordingly.
> >
> > 9) The reviewer is right. It is possible to identify the process $X_{t}^{H}-E\left[ X_{t}^{H}\right], 0\leq t\leq 1,$ linear isometrically with a real valued Gaussian process on the parameter space $B_{1}(0)\times
> > \left[ 0,1\right] $, where $B_{1}(0)$ is the unit ball in $\mathbb{R}^{d}$.
> > However, it seems difficult to us to obtain a sharper bound than in Thm3 by
> > using such a representation.
> >
> > 10) See answer #3 regarding the lower bound. This can be extended here as well (also using answer #7).
> >
> > 11) Thank you for your input. We agree that there is some tradeoff between exploitation (of the gradient) and exploration (through noise). We are more than happy to add a comment on this. Could you please clarify what specifically is overclaimed in the comment about the space-filling property?
> >
> > 12) As mentioned in Remark 3, we cannot any longer exploit
> > the stationarity property of the process in the proofs, if the
> > coefficients are time-inhomogeneous. However, other arguments of the proofs of
> > Thm 3 and Thm 4 can be still employed. In Thm 4 one will get e.g. a similar
> > condition in terms of matrix entries $a_{ij}(t)$ given by a (lengthy)
> > expression depending on the fundamental solution $\Phi (t)$, $A(t)$ and $\Sigma (t)$. In addition other conditions are needed, which in fact question
> > the applicability of such a result in this case.
> >
> > 13) The result we mention in Remark 4 only applies to fractional Brownian motion (fBM) alone, i.e. without any drift term. In contrast, we address the case of a fractional Ornstein–Uhlenbeck process (fOUP), which corresponds to gradient flow where the drift term is a gradient of a quadratic potential, while the noise term is fBM. In addition, the bound stated in Remark 4 contains some additional unknown constants depending on H, which therefore does not precisely capture the dependency on the parameter H (the most important parameter in our analysis).

---

> > > ### Comment · Reviewer_zaJL · 2022-08-09
> > > **Thank you for the response**
> > >
> > > Some of my concerns are addressed. The major issue is in the high-dim analysis, which does not make use of the right geometry and seems to be artificial in many aspects. I will raise my score to 5 but cannot give a higher score due to the mentioned limitations of this work.
> > >
> > > 1. Thanks. Could add these discussion to main text in the future revision.
> > > 2. note that $\\int_{0}^1\\sqrt{\\log\\frac{1}{\\eta}} d \\eta = \\sqrt{\\pi} / 2$
> > > 3. Thanks.
> > > 4. In the regime $1/4 < |\\beta| < 1$, is your bound still tight in $H$? If not perhaps it is more accurate to ease claim.
> > > 5. Thanks.
> > > 6. OK.
> > > 7. Thanks. I recommend to revise theorems to include at least isotropic diffusion matrix, $\\sigma= \\lambda I$ for scalar $\\lambda$, in their future revision.
> > > 8. The definition of $C\_2$ and assumption of $C_1$ look artificial to me: they seem purely rooted from the analysis. I suspect this is because the loose of analysis (as discussed in 9).
> > > 9. Thanks for acknowledging this. This is a weakness of current analysis for high-dim.
> > > 10. OK.
> > > 11. Spacing-fill is fine. I only wanted to point out that spacing-fill might not aways be a desired property, because this hinders the convergence.
> > > 12. I see. It might be worth to remark this in the main text in the future revision.
> > > 13. OK.

---

> > > > ### Author Response · Authors · 2022-08-09
> > > > **Response**
> > > >
> > > > Dear reviewer,
> > > >
> > > > Thank you for your feedback, this is sincerely greatly appreciated.
> > > >
> > > > Regarding your question in point 4, yes the bound appears to be tight experimentally, see the newly added experiment in section B.1.
> > > >
> > > > We will follow your suggestions regarding the other comments your made, thank you.
> > > >
> > > > Best regards,
> > > > The authors

---

> ### Author Response · Authors · 2022-08-08
> **Gentle reminder for response**
>
> Dear Reviewer,
>
> We would like to sincerely thank you for your time and efforts in reviewing our manuscript. We have provided answers to your comments and questions. As the author-reviewer discussion period is ending soon, we would like to kindly request to please let us know if you have any further concerns.
>
> Best regards,
> The authors

---

### Meta-Review · Area_Chair_41Mx · 2022-08-21

**Recommendation:** Accept
**Confidence:** Less certain

**Metareview:**

This paper investigates the use of non-Gaussian (specifically "fractional Brownian") noise in SDEs.  The reviewers found a variety of weaknesses, but overall were positive; I point in particular to the many valuable comments left by reviewer XeCx.  As such, I mark this paper as accept, though I strongly urge the authors to further refine their work based on the copious review feedback below.

**Award:**

No

---

### Decision · Program_Chairs · 2022-09-14

Accept